# Effects of dual-task training on cognitive-motor learning and cortical activation: A non-randomized clinical trial in healthy young adults

Swati M. Surkar[1]*, Chia-Cheng Lin[1], Brittany Trotter[2], Tyler Phinizy[3], Brian Sylcott[4]

**1** Department of Physical Therapy, East Carolina University, Greenville, North Carolina, United States of America, **2** Department of Kinesiology, East Carolina University, Greenville, North Carolina, United States of America, **3** Division of Physical Therapy, University of North Carolina at Chapel Hill, Chapel Hill, North Carolina, United States of America, **4** Department of Engineering, East Carolina University, Greenville, North Carolina, United States of America

* surkars19@ecu.edu

## Abstract

Dual-task (DT) training, which involves the simultaneous execution of cognitive and motor tasks, has been shown to influence task performance and cortical activation, yet evidence on the effects of DT training and cortical activation for complex postural control tasks remains limited. This study investigated the immediate and retention effects of a one-week DT training program on DT learning, performance in DT and single-task conditions, and activation in bilateral prefrontal (PFC) and vestibular cortices in healthy young adults. Eighteen individuals (age = 22.39 ± 1.73 years) participated in the study. The DT paradigm involved a dynamic stability platform (motor task) paired with either a simple or complex auditory reaction time (RT) task (cognitive). Participants completed 20–25 minutes of DT training (18 trials/day) across five consecutive days. DT performance was measured by the duration participants maintained the stability platform within 3 degrees of the horizontal while responding to auditory stimuli. Single-task motor and cognitive performances were also assessed. Cortical activation in the PFC and vestibular cortices was measured using functional near infrared spectroscopy (fNIRS), tracking changes in oxygenated hemoglobin (HbO) concentrations. Pre-training, post-training, and one-week follow-up testing was conducted. The results demonstrate that DT training significantly improves and retains DT performance, likely due to a reduction in cognitive-motor interference. Additionally, DT training led to decreased activation in the bilateral PFC and vestibular cortices, specifically for complex DT condition, suggesting enhanced attentional resource allocation and optimized vestibular input processing, indicative of neural efficiency. Notably, these training effects also transferred to single-task cognitive and motor performances, with corresponding reductions in PFC and vestibular cortex activation, despite the lack of direct training on these tasks. This study advances our

**Data availability statement:** Data is available in supporting information 3 (s3).

**Funding:** The study was funded by ECU start-up funds to Swati M. Surkar.

**Competing interests:** The authors have declared that no competing interests exist.

understanding of the neural mechanisms underlying DT training and underscores the critical role of practice in optimizing cognitive-motor efficiency.

## Introduction

Many daily activities require the simultaneous performance of cognitive and motor tasks, often referred to as dual-tasking. Dual-task (DT) is defined as concurrent execution of cognitive and motor tasks, that can be performed independently and have distinct and separate goals [1,2]. However, the simultaneous performance of these tasks often leads to a decline in one or both tasks' performance, a phenomenon known as DT interference [3,4]. Theoretical models, such as the bottleneck and central sharing models, suggest that limited central processing capacity or competing cognitive-motor resources contribute to DT interference, thereby affecting functional task performance [5–7]. This interference is particularly evident in balance and gait postural control paradigms and has been documented across various populations, including individuals with neurological conditions such as stroke [8,9], cerebral palsy [10], Parkinson's disease [11], and multiple sclerosis [12].

Dynamic balance, a critical component of postural control, is essential for functional tasks such as standing, walking, and mobility. Dynamic balance requires integration of visual, vestibular, and somatosensory inputs, and is mediated by the basal ganglia-cortical loop for higher level, controlled processing and by the brainstem for lower level, automatic processing [13–16]. Dynamic balance is often impaired in individuals with neurological disorders where injury to the central nervous system affects motor and cognitive processing. This disruption can impede both higher and lower-level information processing, leading to a loss of automatic control of dynamic balance. Furthermore, studies have shown that impaired motor systems require increased attentional resources for movement control, thereby exacerbating cognitive-motor interference and making automatic postural control more attention-demanding [17]. Specifically, the addition of a cognitive task to postural control tasks has been shown to influence posture in healthy individuals [18–22], aging populations [23–27] and those with pathological conditions [28–34]. Collectively, the increased attentional demands for controlling dynamic balance heighten the risk of postural instability, loss of balance, and impaired mobility, underscoring the need for targeted rehabilitation training.

While several studies have demonstrated DT interference in healthy as well as neurological populations, very few studies have investigated the effects of DT training in reducing DT interference and enhancing DT performance. Of the few studies that have assessed the effects of DT postural stability training in acute stroke [35], chronic stroke [36,37], healthy elderly [38–41], and young adult [20] populations have shown significant enhancement of postural stability following DT training. However, while the behavioral effects of DT training are recognized, research investigating the neural mechanisms driving these improvements and cortical adaptations with DT training are less understood.

Neuroimaging studies, particularly those employing functional near-infrared spectroscopy (fNIRS), have consistently demonstrated that DT elicits heightened activation across the prefrontal, temporal, and parietal cortices [42–49]. The prefrontal cortex (PFC), in particular, has been implicated in modulating postural control during single-task dynamic balance scenarios [50–52] and exhibits even greater activation during DT dynamic balance tasks [53,54]. Similarly, the vestibular cortices also show increased engagement under DT balance conditions [55–58]. These cortical activations are well-aligned with resource-sharing theory, which underscores the dynamic yet finite nature of attentional resources allocated between cognitive and motor domains, thereby amplifying task interference under DT demands [59,60]. Emerging evidence suggest complex dynamic balance training induces structural and functional cortical adaptations, including gray matter alterations in the PFC and parietal cortices and white matter changes in the supplementary motor and medial parietal cortices [61,62]. Furthermore, preliminary studies in individuals with mild cognitive impairment indicate that DT walking training reduces PFC and motor-associated cortical activation [63,64], consistent with neural efficiency hypothesis. This hypothesis posits that as individuals acquire task proficiency, cortical activation diminishes for equivalent task demands, reflecting optimized neural processing and decreased motor or cognitive load [65]. Despite these advances, the extent to which DT training facilitates cortical adaptations remain poorly understood. Elucidating the neural mechanisms underpinning such training paradigms may offer critical insights into targeted intervention strategy at cortical level.

The objectives of this study were to investigate the effects of one week of DT complex dynamic balance training on (1) DT learning, (2) DT and single-task performance of cognitive and motor tasks, and (3) the activation of PFC and vestibular cortices in healthy young adults. We hypothesized that DT training would optimize activation within the bilateral PFC and vestibular cortices and reduce cognitive-motor interference, thereby enhancing both DT and single-task performance. We assessed cortical activation using functional near infrared spectroscopy (fNIRS), a non-invasive neuroimaging technique, that utilizes specific wavelength of infrared light to estimate the absorption of oxygenated (HbO) and deoxygenated hemoglobin (Hb) within the underlying cortical tissues. A greater concentration of HbO corresponds to a greater amount of activity in the underlying neural tissues [66]. We employed a dynamic balance task, standing on a stability platform, as the motor component of the DT. The dynamic balance task is an ecologically valid motor learning task that engages vestibular, visual, motor, somatosensory, and cognitive systems, serving as an effective measure of the motor learning response to the training intervention [61]. An auditory reaction time (RT) task was chosen as the cognitive component of the DT [67]. We chose to test healthy young adults to determine whether benefits occur within a healthy nervous system, providing the foundation for future studies in neurologic populations. This study will provide valuable insights into whether improvements in DT performance with training are accompanied by neuroplastic changes at the cortical level, laying the groundwork for future research in neurological populations.

## Materials and methods

### Trial design

This study was a prospective, single-group intervention trial with a repeated-measures design. The study consisted of eight total visits. Visits 1, 7, and 8 included assessments of pre-, post-, and follow-up training performance on motor, cognitive, and DT activities. Prefrontal and vestibular cortical activation were measured during the performance of these tasks. Visits 2–6 involved DT training in healthy young adults. All participants provided written informed consent and were compensated for their time and effort. The study was approved by the University and Medical Center Institutional Review Board (UMCIRB) at East Carolina University and was registered at clinicaltrials.gov (NCT04666181). The study was conducted between January 2021 – September 2022.

### Participants

Eighteen healthy young adults were recruited in this study. Inclusion criteria were healthy, young adults in the age range of 18–40 years with intact cognitive and motor functions. Exclusion criteria were: (1) a history of any neurological condition, attention deficit disorder, attention deficit hyperactivity disorder, depression, bipolar disorder; (2) history of concussion;

(3) impaired vision; (4) impaired hearing; (5) balance and vestibular disorders; (6) presence of lower extremity condition, injury, or surgery within the last 3 months; (7) any cognitive, sensory, and communication problem that might prevent completion of the study; (8) current substance abuse or dependence; (9) current use of medications such as selective serotonin reuptake inhibitors that could decrease nervous system excitability; and (10) known cardio-respiratory dysfunctions, all of which could compromise DT training. All the participants were either undergraduate or graduate level students involved in either light, moderate or vigorous physical activities.

A total of 42 participants were assessed for eligibility, with 18 enrolled in the study. As per the CONSORT diagram (Fig 1), all 18 participants were included in the final analyses.

A priori power analysis was conducted using G*Power statistical software [68], informed by our previous experimental work utilizing dynamic stability platform data in healthy young adults [69]. Based on an effect size of 1.2, with the goal of detecting a mean difference in change score of ≥ 3 seconds and a standard deviation of change scores of 2.5 seconds, a sample size of 15 participants was calculated to achieve 95% power at an alpha level of 0.01. To accommodate a potential 20% dropout rate, the final sample size was increased to 18 participants.

## Experimental procedure

The experimental procedure is depicted in Fig 2. The study comprised a total of eight visits. Visit 1 took place on the weekday prior to the onset of training. Visits 2–6 involved DT training and were conducted on consecutive weekdays. Visit 7 occurred on the next weekday following the conclusion of training and Visit 8 was conducted one week after Visit 7 to assess retention effects.

During Visit 1, participants provided demographic information, including height and weight, for the calculation of body mass index (BMI), along with their health history and physical activity levels. Informed consent was obtained from all participants prior to their enrollment and baseline testing. Participants were assigned to a single group for the DT training intervention. All the testing and training were performed in the Pediatric Assessment and Rehabilitation Lab and Sensorimotor Testing and Rehabilitation Lab at East Carolina University.

## Auditory reaction time task

An RT task was utilized as the cognitive component of the DT paradigm, implemented via a custom program and instrument (LabVIEW v.2015, National Instruments). Participants responded to auditory stimuli using handheld clicker remotes, with instructions to press the remotes as quickly as possible upon hearing the tone. The task was designed with two levels of complexity: simple and complex.

In the simple reaction time task (SRT), participants used a single clicker held in their right (dominant) hand to respond to a 500 Hz tone delivered at 60% volume through in-ear headphones. In the complex reaction time task (CRT), participants responded to two distinct tones. The 500 Hz tone was associated with the clicker in the right hand, while a 1000 Hz tone was paired with a separate clicker held in the left (non-dominant) hand. This approach aligns with methodologies used in similar DT studies [26,67]. CRT necessitate the coordination of several cognitive functions, including working memory to retain sound patterns and their corresponding responses, sustained attention to maintain focus, auditory discrimination to identify pitch variations, decision-making to select the appropriate response, and executive action to execute the motor response [70]. The PFC is critically engaged during this multifaceted process. A growing body of research indicates that PFC activation is significantly heightened during high-demand tasks compared to low-demand counterparts [45,70,71]. This augmented neural activity is often accompanied by an increase in cerebral blood volume, which serves as a proxy for neuronal activation and reflects the metabolic demands associated with task performance.

Participants underwent practice sessions for both task conditions prior to data collection, during which they were familiarized with the tones to ensure comprehension of the task requirements.

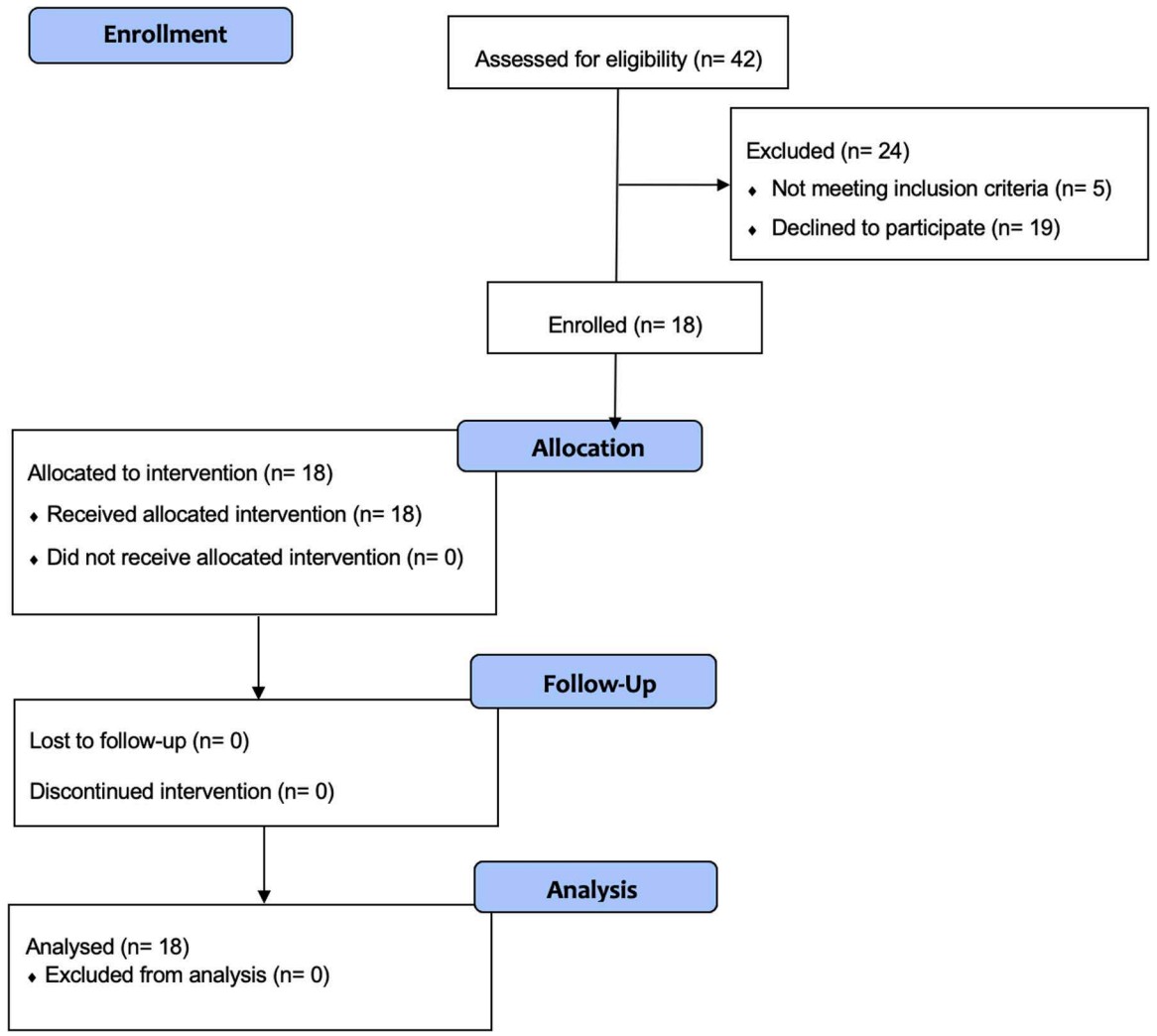

**Fig 1. CONSORT flow diagram of participant enrollment and retention.** *CONSORT flow diagram detailing participant enrollment, allocation, intervention, follow-up, and analysis. A total of 42 participants were assessed for eligibility, with 18 meeting the inclusion criteria and completing the study. The diagram illustrates the process from initial assessment through to the final analysis, including reasons for exclusion and participant retention throughout the study phases.*

### Dynamic balance task

Balance training and testing were conducted using a stability platform (Lafayette Instrument, model 16030L, Lafayette, IN) with a maximum deviation of 30°. Participants were instructed to stand on the platform with their feet facing forward and to maintain the platform in a horizontal position (±3° of level) for as long as possible during 30-second trials. Participants were allowed to hold onto the handrail during initial positioning and rest break but not during testing and training trials. Following a discovery learning approach, participants were encouraged to explore and adopt optimal postural strategies to keep the platform horizontal. Performance for each trial was assessed by measuring the duration (in seconds) that the participant maintained the platform within ±3° of horizontal, as recorded by the PsymLab Psychomotor Control Software (Lafayette Instrument, Lafayette, IN). After each trial, participants received feedback on their performance, specifically the time (in seconds) they kept the platform within the central ±3° range. Participants were encouraged to enhance their balance performance by using various postural strategies. During each visit, balance performance for each trial was

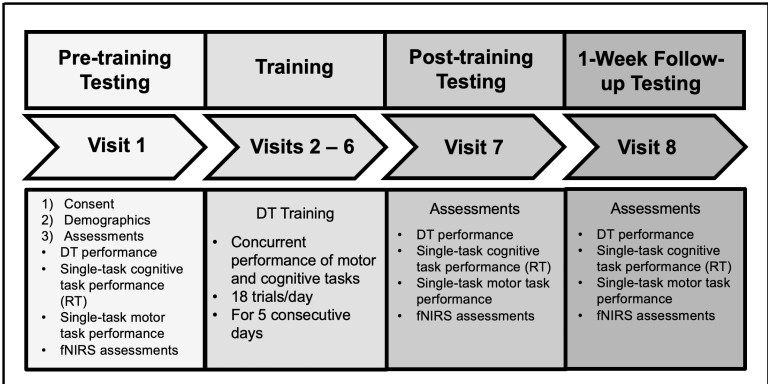

**Fig 2. Experimental procedure timeline and sequence of visits for dual-task training.** *Schematics of the study protocol. The experimental design includes pre-training, post-training, and follow-up assessments of dual-task (DT) performance, single-task cognitive performance (measured as reaction time, RT), and single-task motor performance (measured as balance time in seconds). Functional near-infrared spectroscopy (fNIRS) was utilized to measure cortical activation in the bilateral prefrontal and vestibular cortices. The DT training protocol involved the simultaneous execution of cognitive tasks (simple and complex auditory RT) and motor tasks (balance) for 20–25 minutes per day across five consecutive days. Participants completed 18 DT trials each day, with each trial consisting of 30 seconds of task performance followed by 30 seconds of rest.*

averaged to serve as a measure of balance task performance for that visit. For motor task performance testing, participants performed six trials.

## Dual-task testing and training

The DT required simultaneous performance of a balance task and either SRT or CRT. Testing was conducted at three time points: pre-, post-, and follow-up on visits 1, 7, and 8, respectively.

The DT performance testing included a total of 6 trials. The first 3 trials included concurrent performance of the SRT and dynamic stability task. Whereas the later 3 trials (trials 4–6) included concurrent performance of the CRT task along with the dynamic stability task. Each trial was 30-second in duration followed by 30-second of rest (i.e., quiet standing on the platform while holding onto the handrails). After each trial, verbal feedback of the time spent in balance (s) for that trial was provided so that participants had to employ a trial-and-error system of postural strategies to improve balance task performance [61].

Participants received the DT training intervention on visits 2–6, which spanned 5 consecutive weekdays. There were 18 trials/day during training visits (total 90 trials across 5 training days), and task complexity of the auditory RT task was randomized equally among 18 trials. Participants were allowed 2 minutes of rest after 6 trials to minimize fatigue. Each training visit lasted approximately 20–25 minutes. After each trial, feedback was given to the participants about their performance, and they were encouraged to enhance their performance in the next trial by using various postural strategies.

## Cognitive, motor, and dual-task performance testing

Performance testing on cognitive, motor, and DT occurred on pre-testing (visit 1), post-testing (visit 7), and follow-up testing (visit 8). Cortical activation within PFC and vestibular cortices was concurrently recorded while performing the cognitive, motor, and DT using fNIRS (Fig 3) with a design (A-B1-A-B2-A-B3-A). In this design, cognitive tasks were performed during B1, balance tasks during B2, and DT during B3. Rest periods (A) were incorporated before and after each cognitive, motor and DT testing (Fig 4). Each task and rest trial lasted for 30 seconds. Participants completed a total of 6 trials per visit, consisting of 3 trials for SRT and 3 trials for CRT. Prior to the testing sessions, participants performed three practice trials for both the SRT and CRT while seated to familiarize themselves with the tasks.

## Functional Near-Infrared Spectroscopy (fNIRS) and Data Analysis

During pre-testing, post-testing, and follow-up testing, fNIRS measurements were employed to evaluate cortical activation in the right prefrontal cortex (RPFC), left prefrontal cortex (LPFC), right vestibular cortex (RVEST), and left vestibular cortex (LVEST).

Two sets of 64-channel continuous wave fNIRS (NIRSport, NIRx, Germany) with a total of 16 sources and 16 detectors were used in the tandem model to record the hemodynamic changes in the region of interest (ROI). Two different wavelengths of LED light (760 and 850 nm) were used to detect the changes of both oxy- (HbO) and deoxy-hemoglobin (Hb) in the ROI. The ROIs include the PFC and vestibular cortices in both hemispheres (Fig 3). To standardize the source-detector array for the ROIs, 3 different sizes of caps (EASYCAP, Germany) based on the standard 5–10 layout were used. The NIRS Brain AnalyzIR Toolbox [72] was used for fNIRS signal processing and data analysis which has been described in several publications [58,73]. fNIRS signal processing included converting the raw data of the changes in hemoglobin to optic density and then converted to relative Hb and HbO concentration changes ($\Delta\beta$) via the modified Beer-Lambert Law. The first-level signal processing in subject level used an autoregressive pre-whitening approach with iteratively reweighted least-squares (AR-IRLS) statistical method to eliminate motion artifacts and estimate the mean difference between the baseline and the test conditions [74]. The group level signal processing was performed via a linear mixed effects model [72].

## Outcomes

### Cognitive task performance

Auditory reaction time: Amount of time (ms) to respond to simple or complex auditory tones.

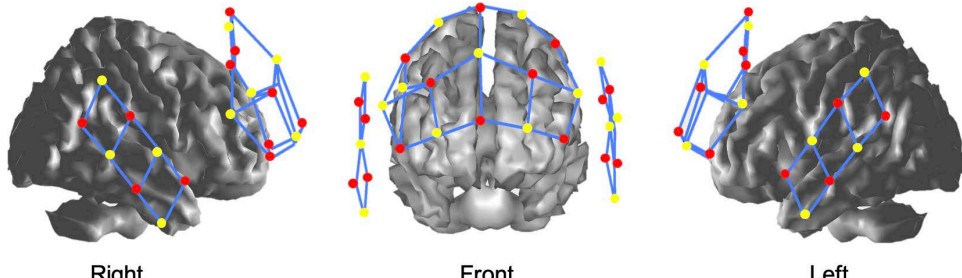

Right Front Left

**Fig 3. fNIRS source-detector array setup. Red Indicates Source; Yellow Indicates Detector.** *This figure illustrates the arrangement of the functional Near-Infrared Spectroscopy (fNIRS) system utilized in the study. The setup features multiple source-detector pairs, with sources indicated in red and detectors in yellow. The sources emit near-infrared light that penetrates the cortical tissues, while the detectors capture the transmitted light to assess changes in cortical oxygenation. The strategic placement and configuration of these components are essential for accurately monitoring cortical activation in the PFC and vestibular cortices during cognitive, motor, and DT conditions.*

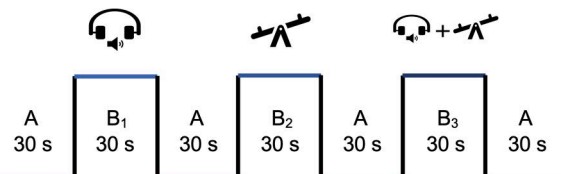

**Fig 4. Study Block Design: Task Structure and Trial Distribution.** *A: 30 seconds rest; $B_1$: auditory RT (cognitive) task (simple RT or complex RT tasks); B2: balance (motor) task; B3: auditory cognitive task + balance tasks (DT). Each participant preformed six trials with three simple RT trials and three complex RT tasks.*

**Motor task performance**: Average amount of time that a participant kept the dynamic stability platform within ± 3º of horizontal during 6 trials for testing.

**Dual-task performance**: Average amount of time that a participant kept the dynamic stability platform within ± 3º of horizontal while concurrently responding to either a simple or complex auditory RT tasks.

### Cortical activation:

a. Oxy-hemoglobin (HbO): Average relative $\Delta\beta$ HbO in each ROI across the active period of cognitive, motor, and DT trials. We used average HbO as a marker of regional brain activation.

b. Deoxy-hemoglobin (Hb): Average relative $\Delta\beta$ Hb in each ROI across the active period of cognitive, motor, and DT trials.

### Statistical analyses

Statistical analyses were conducted using SPSS Statistics 28 (Version 28.0, IBM Corporation, Armonk, NY) with two-sided tests and a significance level set at 0.05. Descriptive statistics, including means and standard deviations for continuous variables were calculated for the study sample.

As the data was normally distributed, a repeated measures ANOVA was employed to assess DT learning across training visits 2–6, with time (visits 2–6) and conditions (simple vs. complex RT tasks) as the within-subject factors. Another repeated measures ANOVA was used to evaluate performance differences in DT, cognitive, and motor conditions across pre-, post-, and follow-up testing, with time as the within-subject factor. Significant main effects were examined further using a Bonferroni post-hoc analysis.

For the fNIRS data, t-tests used to examine if the regression coefficient differed from 0 and to compare the difference between different ROIs. The Benjamin-Hochberg procedure was used to adjust the p-value to decrease the false discovery rate (FDR) and Bonferroni correction was used for among visit and task comparison. The significance level for fNIRS comparison was set as FDRp < 0.05 and FDRp < 0.017 for among visit comparison. Results are reported as mean ± standard deviation. P-values less than 0.05 were considered statistically significant.

### Results

Table 1 shows baseline characteristics of the participants included in the study.

There were no adverse events that occurred among the participants included in the study and no participants were withdrawn from the study.

### Effects of Dual-task Training on Dual-Task Learning

Fig 5 illustrates DT performance across five training visits for SRT and CRT (cognitive task). There was a significant main effect of time (p = 0.01), which demonstrates a progressive improvement in DT dynamic balance performance with training

**Table 1. Baseline Characteristics of Study Participants.**

| Participants (n = 18) | |
|---|---|
| **Characteristics** | |
| Age (mean, SD, range) | 22.05 (±1.79), 19–25 years |
| Female/Male | 12/6 |
| Dominant Side (R/L) | R = 18 |
| Weight (kgs) | 71.76 ± 16.13 |
| Height (meters) | 1.73 ± 0.11 |
| BMI (kg/ (m*m)) | 23.76 ± 3.85 |

for both cognitive conditions. Post-hoc analysis reveals significant differences in performance across visits for both task types (Table 2). There was no significant time x condition interaction (p = 0.713), which indicates that the improvements in DT performance were consistent across simple and complex RT. Overall, these results highlight a progressive enhancement in DT learning following 1 week of training in healthy young adults.

### Dual-task Performance

There was a significant main effect of time (p = 0.001) indicating improved DT performance from pre- to post-training and sustained improvements at the follow-up visit (p = 0.001) for both task conditions. Post-hoc analyses revealed significant differences between pre- and post-training, as well as between pre-training and follow-up testing, demonstrating that the DT training resulted in significant performance enhancements that were maintained one week after the training (Fig 6).

### Cognitive task performance

Median RT for both simple and complex conditions exhibited a significant main effect of time (p = 0.001). Post-hoc analyses reveal that RT for both conditions decreased from pre- to post-training and were sustained at the 1-week follow-up (p < 0.001). RT was longer for CRT compared to SRT, reflecting increased information processing time with

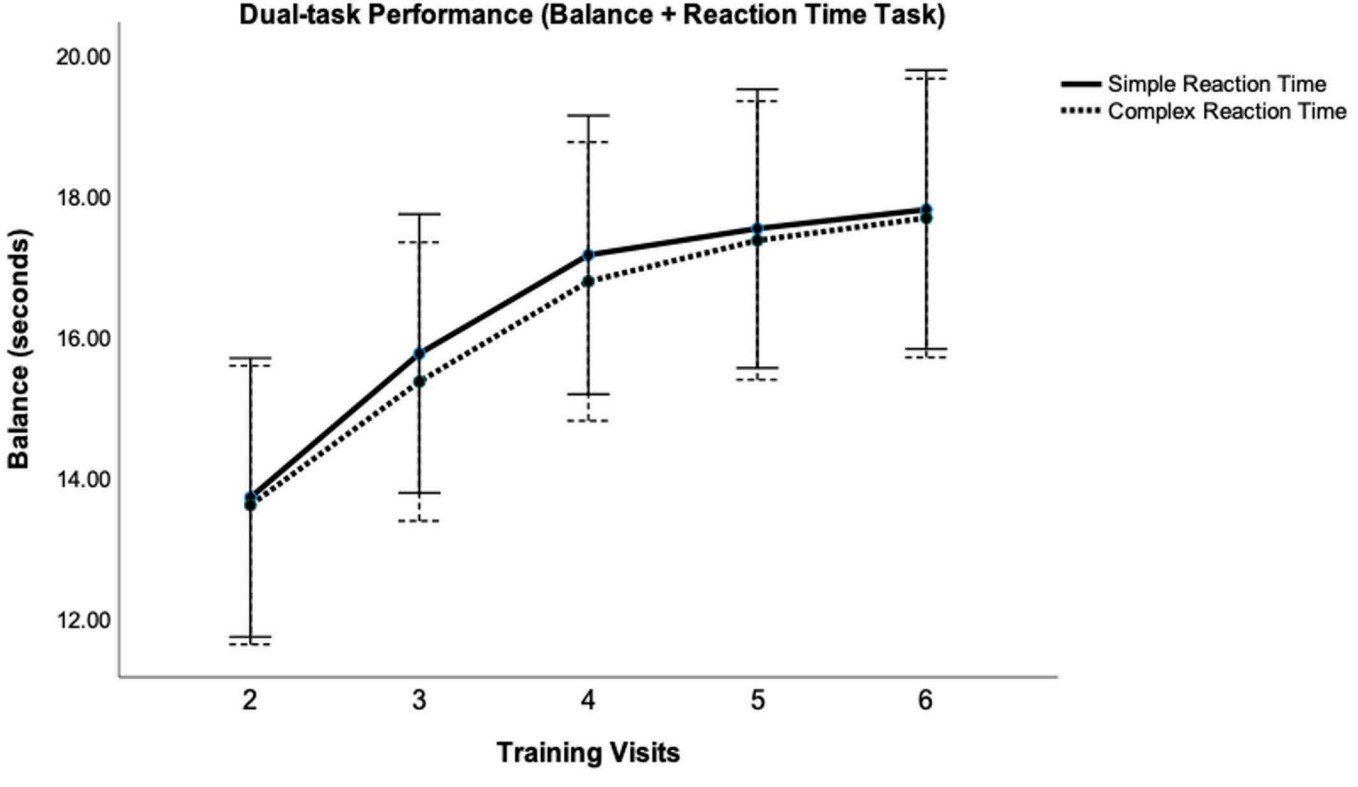

**Fig 5. Dual-task learning across training visits for simple and complex reaction times.** *Dual-task (DT) performance during SRT and CRT across five training visits. Participants were required to maintain the stability platform within 3 degrees of horizontal while simultaneously responding to SRT and CRT. DT performance, indicated by the average number of seconds in balance during 30-second trials, progressively improved over the course of the five training visits (Visits 2–6) for both cognitive conditions. Data points represent the mean performance across 18 trials, and error bars denote the 95% confidence interval.*

**Table 2. Post-hoc analysis for visit-wise differences in dual-task learning. Significant differences between visits are indicated with an asterisk (*).**

**Dual task performance (simple reaction time)**

| Visit vs. | Visit | Significance | Mean Difference | Error of Difference | Lower Bound | Upper |
|---|---|---|---|---|---|---|
| 2 | 3 | 0.14 | −2.04 | 1.38 | −4.79 | 0.7 |
| 2 | 4 | 0.015* | −3.44 | 1.38 | −6.19 | −0.69 |
| 2 | 5 | 0.007* | −3.82 | 1.38 | −6.56 | −1.07 |
| 2 | 6 | 0.004* | −4.08 | 1.38 | −6.83 | −1.34 |

Dual task performance (complex reaction time)

| Visit vs. | Visit | Significance | Mean Difference | Error of Difference | Lower Bound | Upper |
|---|---|---|---|---|---|---|
| 2 | 3 | 0.231 | −1.75 | 1.45 | −4.64 | 1.14 |
| 2 | 4 | 0.032* | −3.18 | 1.45 | −6.06 | −0.29 |
| 2 | 5 | 0.011* | −3.79 | 1.45 | −6.64 | −0.87 |
| 2 | 6 | 0.006* | −4.08 | 1.45 | −6.96 | −1.19 |

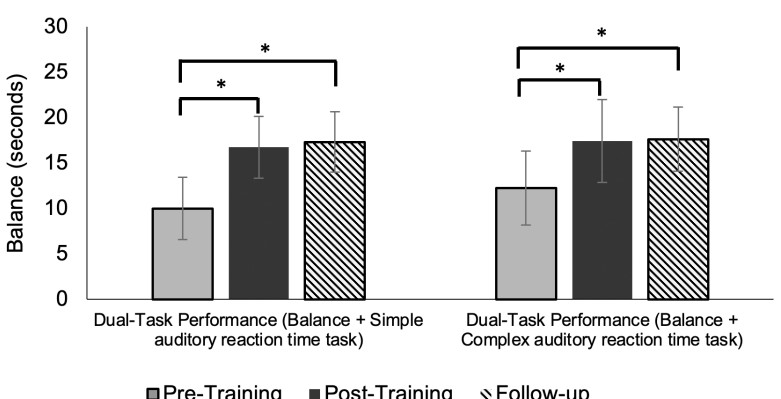

**Fig 6. Dual-task (DT) performance during simple and complex cognitive conditions.** *DT performance during pre-, post, and follow-up training visits. DT performance improved and retained with training. The bars denote the mean and error bars denotes standard deviation.*

greater cognitive task complexity. Despite this, DT training resulted in improvements and retention of cognitive performance across both task conditions (Fig 7).

## Motor task performance

A significant main effect of time (p = 0.001) indicates notable improvements in balance performance following DT training. Post-hoc analysis reveals significant differences in performance from pre- to post-training (p = 0.001) and from pre-training to follow-up (p = 0.001). These results demonstrate that DT training enhances motor task performance, with improvements retained for up to one week (Fig 8).

## Cortical activation:

**Cognitive tasks.** *Prefrontal Cortices (PFC) Activation.* There was a significant main effect (p < 0.01) of time (pre-, post-, and follow-up testing) and time x condition (simple and complex RT) interaction for bilateral PFC activation.

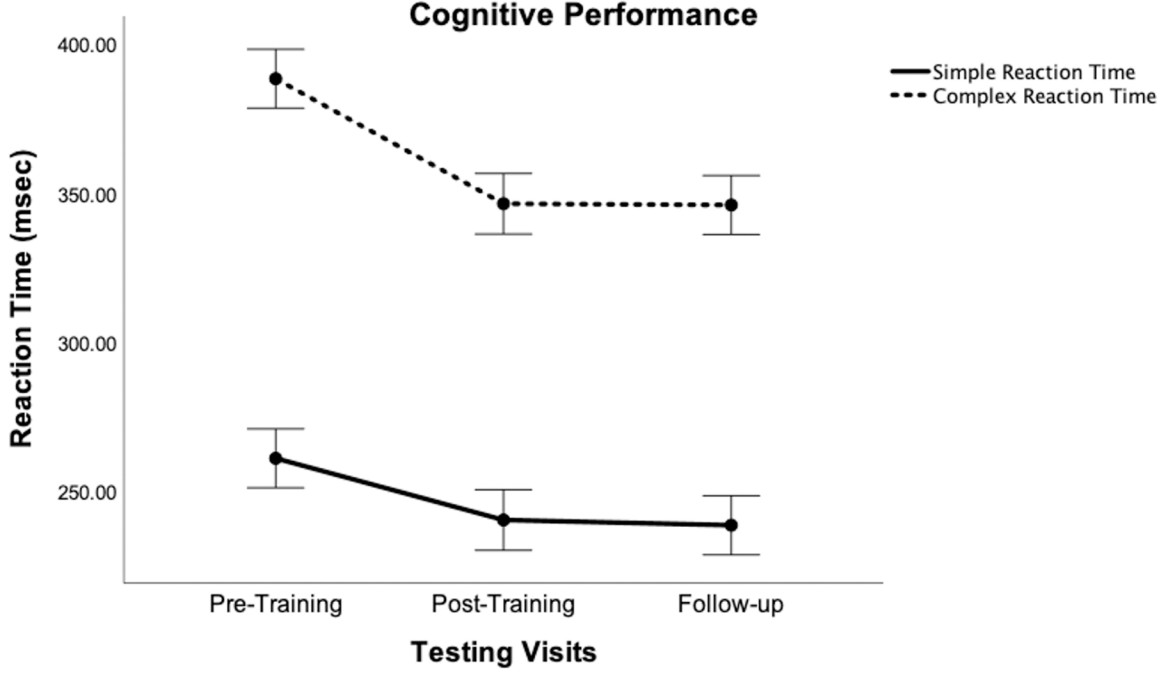

**Fig 7. Median reaction time (RT) across pre-, post-, and follow-up testing for simple and complex conditions.** *RT decreased and retained with training for simple as well as complex cognitive conditions, which indicates faster information processing time.*

Post-hoc analyses suggest that the RPFC activation significantly decreased during simple as well as complex RT tasks in the post- (FDRp < 0.001) and follow-up testing (FDRp < 0.001) compared to pre-testing (Fig 9 A and C). The LPFC only exhibited significantly increased activation during the simple RT tasks in the follow-up visit compared with pre- (FDRp < 0.001) and post-testing (FDRp < 0.001) (Fig 9 A). These results indicate DT training-related optimization primarily in RPFC activation during simple and complex cognitive tasks.

***Vestibular cortices activation.***There was a significant main effect (p < 0.01) of time (pre-, post-, and follow-up testing) and time x condition (simple and complex RT) interaction for bilateral vestibular activation.

Post-hoc analyses suggest that the bilateral vestibular cortices activation significantly decreased during simple as well as complex RT tasks in the post- (FDRp < 0.001) and follow-up testing (FDRp < 0.001) compared to pre-testing (Fig 9 B and D). These results indicate DT training-related optimization in bilateral vestibular cortices activation during simple as well as complex cognitive tasks.

**Balance tasks.** There was a significant main effect (p < 0.01) of time (pre-, post-, and follow-up testing) for bilateral PFC and vestibular cortices activation for the balance (motor) task. Post-hoc analyses revealed a significant decrease in activation within the bilateral PFC and vestibular cortices during post-testing (FDRp < 0.001) and follow-up testing (FDRp < 0.001) compared to pre-testing (Fig 9 E and F). These results indicate DT training-related optimization in bilateral PFC and vestibular cortices activation during the single (motor *only)* task.

**Dual-tasks.** There was a significant main effect (p < 0.01) of time (pre-, post-, and follow-up testing) and time x condition (simple and complex RT) interaction for bilateral PFC and vestibular cortices activation during DT.

## PFC and vestibular cortices activation during simple RT dual-task

Post-hoc analyses revealed significantly increased activation in the right PFC during post-testing in the simple RT DT condition, compared to both pre- (FDRp < 0.001) and follow-up (FDRp < 0.001) testing. No significant differences (p > 0.05)

## Motor Task Performace

**Fig 8. Motor performance across pre-, post-, and follow-up testing for simple and complex cognitive conditions.** Balance time increased and retained with training.

were found between the pre- and follow-up testing (Fig 9 G). The left PFC demonstrated a significant decrease in activation in the follow-up testing (FDRp < 0.001) compared to pre- and post-testing during the simple RT DT conditions. The right vestibular and left vestibular cortices had a similar activation pattern during the simple RT DT condition, in which cortical activation significantly decreased through the post- and follow-up testing (all FDRp < 0.001) compared to pre-testing (Fig 9 H). These results indicate DT training-related optimization primarily in bilateral vestibular cortex activation.

### PFC and vestibular cortices activation during complex RT dual-task

There was a significant decrease in bilateral PFC and right vestibular cortices activation from pre- to post- (FDRp < 0.001) and follow-up testing (FDRp < 0.001) during the complex RT DT condition (Fig 9 I and J). The left vestibular cortex showed a significant decrease in activation from pre- to post-testing (FDRp < 0.001) but an increase in activation during follow-up testing (FDRp < 0.001) compared to post-testing (Fig 9 J). These results indicate DT training-related optimization in bilateral PFC and right vestibular cortex activation.

## Discussion

To the best of our knowledge, this is the first time the effects of DT training on DT performance and retention, and the associated cortical activation were investigated. Our results indicate that DT training plausibly— 1) enhances DT performance and retention, 2) optimizes bilateral PFC and vestibular cortex activation, and 3) translates to single motor task and single cognitive task performance and optimizes associated PFC and vestibular cortex activation. These noteworthy findings provide impactful insights into the potential impact of DT training on performance, cortical activation, and the underlying mechanisms.

### Enhancement and retention of dual-task performance

The DT in this study consisted of maintaining dynamic balance while responding to simple and complex auditory RT tasks. The DT performance was quantified as the average amount of time a participant kept the stability platform within ± 3º of horizontal while simultaneously performing the auditory tasks. After five days of DT training, DT performance enhanced during simple as well as complex cognitive task conditions. Our findings are in agreement with other studies that have employed similar DT paradigms in healthy, aging, and neurological populations and demonstrated improvement in DT

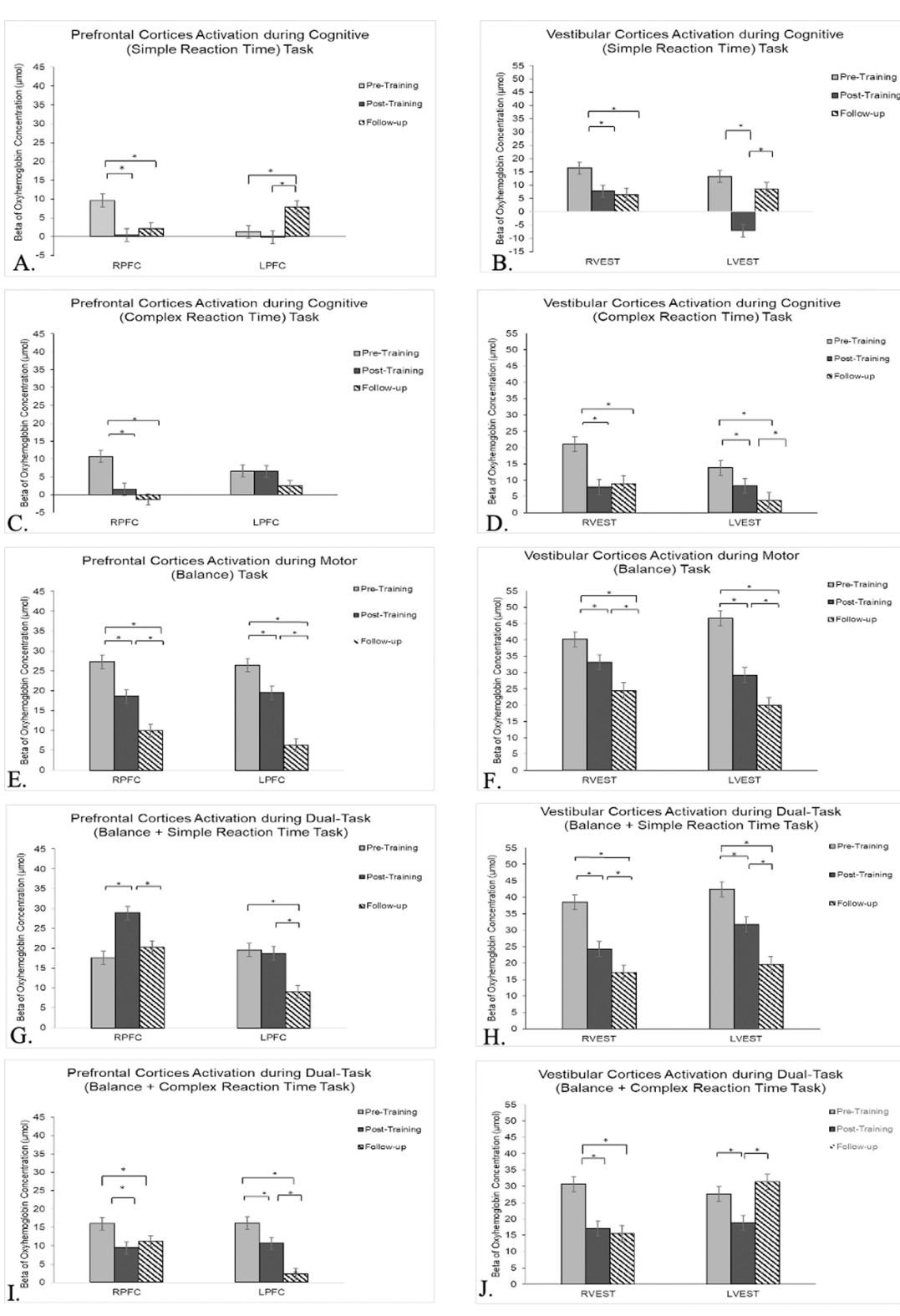

**Fig 9. A)** Prefrontal cortex (PFC) activation during simple reaction time task (SRT); **B)** Vestibular cortex (VEST) activation during SRT; **C)** PFC activation during complex reaction time (CRT); **D)** VEST activation during CRT; **E)** PFC activation during balance task; **F)** VEST activation during balance task; **G)** PFC activation during SRT dual-task; **H)** VEST activation during SRT dual-task; **I)** PFC activation during CRT dual-task; **J)** VEST activation during CRT dual-task; * FDRp<0.001.

performance on postural stability tasks [34]. The progressive increase in DT performance and retention across training days suggests effective motor learning through repetitive practice, which potentially leads to reduced cognitive-motor interference. The potential mechanisms for these improvements could be because of a) effective allocation of attentional resources– training might have enhanced participants' ability to effectively allocate attentional resources between balance and cognitive tasks. Furthermore, participants might have learned optimal strategies by shifting attention to an external focus that improved DT performance. b) Automaticity of motor tasks— performing the balance task might be more attention demanding at the beginning; however, DT training likely facilitated the transition of this task to a more automatic process. This reduction in attentional demand could have freed cognitive resources, reducing cognitive-motor interference [75]. c) Motor planning shifts— with DT training, the motor task became automatic by shifting motor planning and executive control from higher, cognitive centers to more automatic, lower-level processes [76]. Lastly, the use of an overhead harness likely provided a safe environment, which may have facilitated learning by reducing safety concerns and allowing participants to focus more on task performance.

## Optimization of PFC activation during dual-task

The PFC is crucial for allocating attentional resources and processing information required for postural control [52]. Prior studies have shown an increase in the PFC activation in response to postural control tasks in healthy individuals as well as in individuals with neurological disorders [51,77–80]. Specifically, activation in the PFC may reflect attentional processing in the regulation of postural control. Therefore, postural control tasks that demand attention have shown to increase the activation within the PFC. To that extent, the DT paradigms have been shown to increase PFC activation due to greater reliance on attentional resources while simultaneously processing the cognitive demands and maintaining postural control [48,80–82]. Our study results suggest that after DT training, bilateral PFC activation decreased immediately following training, and the effects were retained through the 1-week follow-up visit. The decrease in PFC activation following DT training was significant when the postural control task was combined with a CRT task as compared to the SRT task. Decrease in PFC activation following DT training may be due to practice induced neural efficiency. Specifically, post-training reduction in PFC could be attributed to automaticity of the complex dynamic stability task combined with the CRT task that lowered attentional demands and subsequently required fewer neural resources in attentional network. Furthermore, retention of optimization of PFC activation until 1 week after DT training indicates the retention of neural efficiency following DT training. Our study results contradict the findings of Bohlke et al., (2023) who didn't exhibit changes in PFC activation following DT training [83]. However, the discrepancy can be explained by different intervention protocols between the two investigations. Our study findings are in agreement with other DT training studies in individuals with mild cognitive impairments [64] and in older adults [84], skill acquisition after DT training [85], and in DT processing [86]. Surprisingly, variable trends in right and left PFC activation were seen when the dynamic stability task was combined with SRT. There was a greater PFC activation immediately post- DT training while there were no significant differences in left PFC before and after training. We speculate that the SRT might be cognitively less demanding; hence, the participants might have prioritized the motor control task over the cognitive task that required greater attentional resources leading to greater right PFC activation.

## Optimization of vestibular cortex activation during dual-task

Our primary findings indicate that bilateral vestibular cortex activation decreased following DT training for both SRT and CRT, with retention of effects after 1 week. The dynamic balance task in this study involved complex interactions among sensorimotor, proprioceptive, and vestibular inputs. Consequently, the DT in our study placed significant reliance on vestibular inputs. The vestibular cortex regions we identified corresponded to the bilateral temporo-parietal and superior temporal gyri. Evidence suggests that these areas are implicated in the processing of vestibular inputs [56,57]. Thus, the practice of DT over the course of a week may have led to neural efficiency, reflected by the reduced activation within the

bilateral vestibular cortices observed post-training. Although previous studies have explored vestibular cortex activation during balance tasks [56–58], to the best of our knowledge, this is the first study to demonstrate the optimization of vestibular cortex activation with DT training.

## PFC activation during single task cognitive condition

All participants in this study underwent DT training but we also assessed activation within the bilateral PFC and vestibular cortices during the single task cognitive and single task motor conditions. The single task cognitive conditions included simple and complex RT tasks, while the single task motor condition included the dynamic stability task. During the DT condition, the motor task was paired with both SRT and CRT. Our study findings indicate reduction in the right PFC during both SRT as well as CRT following DT training. In our study, SRT and CRT required participants to respond to auditory stimuli of 500 Hz and 1000 Hz respectively. The participants had to respond to the auditory stimuli as soon as they heard the tone. Therefore, the task involved directing attention to an auditory stimuli, which is a primary function of the right PFC [87]. Notably, the practice effects of DT might have transferred to a single-task cognitive condition. Although the single task cognitive condition was not explicitly trained, participants' practice during DT training potentially led to more efficient utilization of PFC resources, resulting in a reduction in right PFC activation. This reduction was observed immediately after DT training as well as after the 1-week follow-up.

## Vestibular cortices activation during single task cognitive condition

Our results indicate that DT practice might have led to decreased activation in the bilateral vestibular cortices during both SRT as well as CRT. Although the connections between the vestibular and cognitive system are not fully understood [88], there is recognized interdependence between these systems [88]. Previous studies have shown that vestibular deficits can lead to cognitive impairments, particularly in attention, memory, and executive function [89]. Anatomically and physiologically, the interaction between the cognitive and vestibular systems is supported by a neuronal projection between the vestibular centers and the cerebral cortex. The vestibular system interacts with various cognitive processes including spatial navigation, spatial perception, body representation, mental imagery, attention, memory, risk perception, and social cognition [90]. In our study, SRT and CRT involved responding to the auditory stimuli, which required attention and memory as well as spatial navigation to respond to different auditory stimuli while maintaining postural control. The reduction in the bilateral vestibular cortices during a single task cognitive condition likely reflects practice-induced neural efficiency and optimized use of vestibular resources for attention, memory, and spatial navigation, which may have resulted to reduction and retention of vestibular cortex activation.

## PFC activation during the single task motor condition

Our study incorporated a dynamic balance training protocol as a motor task, where participants experienced platform perturbations in the mediolateral direction. This task engaged multiple postural control strategies, including somatosensory, proprioceptive, vestibular, and visual-cognitive systems. Previous research employing a similar dynamic stability platform paradigm has demonstrated activation in the bilateral PFC in both healthy individuals and stroke patients when subjects experienced forward and backward perturbations while standing on the platform [52,91]. The activation of the PFC is particularly significant as it underscores its role in executive functions, including directing and maintaining attention and regulating postural control [92]. Neural efficiency with practice might have led to a decrease in PFC activation during the motor task.

## Vestibular cortex activation during the single task motor condition

In this study, the motor task using the dynamic stability platform required heavy reliance on vestibular inputs. Karim et al., 2012 also identified the modulation of cortical activation in the superior temporal gyrus and supramarginal gyrus in

response to various difficulty levels of dynamic balance tasks [57]. The superior temporal gyrus is involved in the integration of auditory and vestibular information, particularly in the perception of spatial orientation and movement [93,94]. This region is connected to the vestibular cortex and plays a role in the perception of motion and the stabilization of visual images during head movements [93]. The supramarginal gyrus is associated with multisensory integration, including vestibular inputs. It is involved in spatial awareness and the perception of body position in space. The supramarginal gyrus has connections to the vestibular cortex and is important for processing vestibular information that contributes to balance and spatial orientation [95]. Neural efficiency with practice might have led to a decrease in vestibular cortex activation during the motor task.

## Study limitations and future research directions

Our study has a few limitations. 1) It's a single group intervention study. Lack of a control group limits the interpretation about causal effects of intervention. 2) Other cortical areas such as premotor, supplementary motor, parietal cortical areas are involved in balance task; however, were not assessed in this study. 3) The study includes healthy young adults; hence, generalization of study results to other populations might be limited.

## Conclusions

Our findings reveal that DT training has potential to enhance DT performance by refining dynamic balance control during concurrent cognitive tasks. The observed improvements across both SRT and CRT suggest that such training alleviates cognitive-motor interference, likely through the development of motor automaticity and more efficient attentional resource allocation. The post-training reduction in bilateral PFC activation highlights a decrease in the cognitive load necessary for maintaining postural stability, indicative of practice-induced neural efficiency. Furthermore, the concurrent reduction in bilateral vestibular cortex activation suggests a more optimized processing of vestibular inputs, which contributes to the observed improvements in balance control. These neural adaptations, which were maintained at a 1-week follow-up, underscore the enduring effects of DT training on cortical function. This study advances our understanding of the neural mechanisms underlying DT training, emphasizing the critical role of practice in achieving cognitive and motor efficiency. Future research should investigate the generalizability of these findings to diverse populations, including older adults and individuals with neurological impairments, and explore the involvement of additional cortical regions, such as the premotor and parietal areas, to develop a more comprehensive model of DT training effects.

## Supporting information

**S1 Checklist. CONSORT 2010 checklist of information to include when reporting a randomized trial.**
(DOCX)

**S2 Appendix. Glossary of abbreviated terms.**
(DOCX)

**S3 Data. Raw and processed data files.**
(XLSX)

**S4 Protocol. Detailed experimental protocol.**
(PDF)

## Acknowledgments

The authors would like to thank the study participants for generously giving their time.

## Author contributions

**Conceptualization:** Swati M. Surkar.

**Data curation:** Swati M. Surkar.

**Formal analysis:** Swati M. Surkar, Chia-Cheng Lin.

**Funding acquisition:** Swati M. Surkar.

**Investigation:** Swati M. Surkar, Chia-Cheng Lin, Brittany Trotter, Tyler Phinizy.

**Methodology:** Swati M. Surkar, Chia-Cheng Lin, Brittany Trotter, Tyler Phinizy.

**Project administration:** Swati M. Surkar, Chia-Cheng Lin.

**Resources:** Swati M. Surkar, Chia-Cheng Lin, Brian Sylcott.

**Software:** Swati M. Surkar, Chia-Cheng Lin, Brian Sylcott.

**Supervision:** Swati M. Surkar, Chia-Cheng Lin.

**Validation:** Swati M. Surkar, Chia-Cheng Lin.

**Visualization:** Swati M. Surkar.

**Writing – original draft:** Swati M. Surkar.

**Writing – review & editing:** Swati M. Surkar, Chia-Cheng Lin, Brittany Trotter, Tyler Phinizy, Brian Sylcott.

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
