## [Decision Letter · Decision Letter 0]

4 Nov 2024

PONE-D-24-40678Effects of dual-task training on cognitive-motor learning and cortical activation: a non-randomized clinical trial in healthy young adultsPLOS ONE

Dear Dr. Surkar, Thank you for submitting your manuscript to PLOS ONE. After careful consideration, we feel that it has merit but does not fully meet PLOS ONE’s publication criteria as it currently stands. Therefore, we invite you to submit a revised version of the manuscript that addresses the points raised during the review process.

After careful consideration by 2 Reviewers and an Academic Editor, all of the critiques of the Reviewers must be addressed in detail in a revision to determine publication status. If you are prepared to undertake the work required, I would be pleased to reconsider my decision, but revision of the original submission without directly addressing the critiques of the Reviewers does not guarantee acceptance for publication in PLOS ONE. If the authors do not feel that the queries can be addressed, please consider submitting to another publication medium. A revised submission will be sent out for re-review. The authors are urged to have the manuscript given a hard copyedit for syntax and grammar.

We look forward to receiving your revised manuscript.

Kind regards,

Stephen D. Ginsberg, Ph.D.

Section Editor

PLOS ONE

4. Thank you for stating the following financial disclosure: “The study was approved by the University and Medical Center Institutional Review Board (UMCIRB) at East Carolina University. The IRB number is 20-001938”.

6. Please ensure that you refer to Figures 7,8 and 9 in your text as, if accepted, production will need this reference to link the reader to the figure.

7. We note that there is identifying data in the Supporting Information file <S3_Data.xlsx>. Due to the inclusion of these potentially identifying data, we have removed this file from your file inventory. Prior to sharing human research participant data, authors should consult with an ethics committee to ensure data are shared in accordance with participant consent and all applicable local laws. Data sharing should never compromise participant privacy. It is therefore not appropriate to publicly share personally identifiable data on human research participants. The following are examples of data that should not be shared: -Name, initials, physical address -Ages more specific than whole numbers -Internet protocol (IP) address -Specific dates (birth dates, death dates, examination dates, etc.) -Contact information such as phone number or email address -Location data -ID numbers that seem specific (long numbers, include initials, titled “Hospital ID”) rather than random (small numbers in numerical order) Data that are not directly identifying may also be inappropriate to share, as in combination they can become identifying. For example, data collected from a small group of participants, vulnerable populations, or private groups should not be shared if they involve indirect identifiers (such as sex, ethnicity, location, etc.) that may risk the identification of study participants. Additional guidance on preparing raw data for publication can be found in our Data Policy (https://journals.plos.org/plosone/s/data-availability#loc-human-research-participant-data-and-other-sensitive-data) and in the following article: http://www.bmj.com/content/340/bmj.c181.long. Please remove or anonymize all personal information (<specific identifying information in file to be removed>), ensure that the data shared are in accordance with participant consent, and re-upload a fully anonymized data set. Please note that spreadsheet columns with personal information must be removed and not hidden as all hidden columns will appear in the published file.

**Comments to the Author**

1. Is the manuscript technically sound, and do the data support the conclusions?

Reviewer #1: Yes

Reviewer #2: Partly

2. Has the statistical analysis been performed appropriately and rigorously? 

Reviewer #1: Yes

Reviewer #2: Yes

3. Have the authors made all data underlying the findings in their manuscript fully available?

Reviewer #1: Yes

Reviewer #2: Yes

4. Is the manuscript presented in an intelligible fashion and written in standard English?

Reviewer #1: Yes

Reviewer #2: Yes

5. Review Comments to the Author

Reviewer #1: The authors performed a study to examine the effect of a DT training on DT learning, DT and single-task, and activation in bilateral prefrontal and vestibular cortices in young adults. The study is well conducted and presented; however, I have a few comments.

Introduction

The introduction establishes dual-task training's importance for motor and cognitive processes but could be strengthened with more emphasis on neural efficiency mechanisms, particularly regarding the cortical areas implicated in dual-task scenarios. Expanding on how attentional resource allocation theories, like resource allocation theory and the neural efficiency hypothesis, might explain cortical adaptations could deepen the reader’s understanding of the neural mechanisms expected in DT performance improvements.

Methods

Please describe how intact cognitive and motor functions were assessed in the sample.

It would be interesting, in addition to the sample morphologic characteristics, to describe their level of cognitive (eg students) and physical activity (sedentary) if you collected them.

Please give an additional explanation of why complex reaction time tasks are more resource-demanding for cortical areas.

Results

Please use international metric Kg and m.

Place the table 1 heading on top of it.

Reviewer #2: The authors present results of a single-arm study assessing 5-days of dual-task training and its potential impact on dual-task and single task performance the day after training ends and 1-week post-training. The authors also assess changes in cortical activation through fNIRS in the prefrontal cortex (PFC) and vestibular cortex. Authors show improvement in balance during training and improvements in task performance (both single and dual-task) immediately post training and at follow-up. There are also changes observed in cortical activation. The manuscript will be strengthened if the authors address the following points.

1. Authors should clarify the meaning of "trials" and "blocks". It seems that the meaning may change in different sections of the paper which causes some confusion. For example, line 201 says 6 trials with one trial per block but line 209 indicates 3 blocks of six trials, Figure 4 refers to each component as a block and line 222 says each trial had a duration of 210 seconds, while a block was 30 seconds.

2. Authors should clarify their analytic approach for the fNIRS data including the multiple comparison correction strategy. Authors state using both FDR and Bonferroni correction. In lines 253-261, authors mention linear mixed effects models, but then later (lines 287) they talk about mixed model ANOVA and then say a p-value less than 0.05 is considered significant, when earlier they have lower cut-offs presented. I wonder if it would make sense to just describe the methods of analysis just in the statistical analysis section...

3. Authors use various forms of ANOVA for analysis. Were the assumptions of these methods met by the data?

4. Authors need to be careful in their statements about their findings. Although they acknowledge in the limitations that there is no control group, which limits interpretation of causality, much of their phrasing throughout the manuscript suggests causality. Authors do not know what might have happened to cortical activation or even performance on tasks at the different time points without the training.

Minor points

1. Line 42: authors state that DT training led to decreased activation in the bilateral PFC, but this was only true (in the dual-task) for balance+complex reaction time.

2. Figure 2: I suggest adding the timing of visits 7 and 8 to the figure (visit 7 the day after training ends and visit 8 1-week later) so that readers know they are not all consecutive days without reading the text.

3. Would the protocol for the reaction time tasks changed if individuals were left-handed? I realize all participants were right-handed in this study, but it might be worth stating that in the description of the reaction time task section (lines 169-181), so a reader knows that (especially if it would affect the protocol if someone was actually left-handed). Currently, the reader doesn't know everyone is right-handed until Table 1.

4. line 221: change "30 second" to "30 seconds"

5. Table 1: I would include the range of ages, even though the data are provided in supplemental material (especially since the inclusion criteria had a much wider age range than what was ultimately used).

6. Table 2 -for the complex reaction time section, change "Time vs Time" to "Visit vs Visit" to be consistent with the simple reaction time section

7. Figure 6: there is a typographical error in the row labels for the balance+SRT (change "Performace" to "Performance"). Also, define the error bars and * in the figure caption. Is the bar height the observed mean at the different time points? This should be clarified. Authors might also consider a more informative plot, especially given the small sample size (box-plots with individual data points overlaid) - and consider that same type of plot for other figures that are presenting observed means (if that is what is presented).

8. lines 351-352 - authors have switched SRT and CRT. RT was longer for CRT not SRT.

9. Figure 8: row labels - change "testing" to "training". Also define the error bars and *.

10. Figure 9 caption H is listed twice - the second one should be "J"

11. All references to figures in lines 374 through the end of the results should be changed from Figure 6 to Figure 9

6. PLOS authors have the option to publish the peer review history of their article (what does this mean? ). If published, this will include your full peer review and any attached files.

**Do you want your identity to be public for this peer review?**  For information about this choice, including consent withdrawal, please see our Privacy Policy .

Reviewer #1: **Yes: ** Maria António Castro

Reviewer #2: No

---

## [Author Response · Author response to Decision Letter 1]

8 Feb 2025

Reviewer #1: The authors performed a study to examine the effect of a DT training on DT learning, DT and single-task, and activation in bilateral prefrontal and vestibular cortices in young adults. The study is well conducted and presented; however, I have a few comments.

Introduction

1. The introduction establishes dual-task training's importance for motor and cognitive processes but could be strengthened with more emphasis on neural efficiency mechanisms, particularly regarding the cortical areas implicated in dual-task scenarios. Expanding on how attentional resource allocation theories, like resource allocation theory and the neural efficiency hypothesis, might explain cortical adaptations could deepen the reader’s understanding of the neural mechanisms expected in DT performance improvements.

Response: Thank you for your valuable suggestions. We agree that incorporating a foundational theoretical perspective on cortical adaptations resulting from dual-task practice, along with relevant neuroimaging literature, would significantly strengthen the introduction section. We have revised the introduction accordingly and incorporated the suggested changes. Please refer to lines 85–103.

Methods

2. Please describe how intact cognitive and motor functions were assessed in the sample.

Response: Cognitive and motor dysfunctions were assessed using a comprehensive screening questionnaire, which included questions related to:

1. A history of any neurological condition, such as attention deficit disorder, attention deficit hyperactivity disorder, depression, bipolar disorder, balance impairment, or vestibular disorder.

2. History of concussion.

3. Impaired vision or hearing.

4. Recent lower extremity injury that could affect balance.

5. Any extremity, soft tissue, orthopedic, or vascular injury (e.g., peripheral vascular disease) that might impact balance.

6. Any cognitive, sensory, or communication problem that could prevent study completion.

7. Current substance abuse or dependence.

8. Current use of medications (e.g., selective serotonin reuptake inhibitors) that could reduce nervous system excitability.

9. Known cardiorespiratory dysfunction.

Additionally, the study sample consisted of undergraduate and graduate students, ensuring intact cognitive function. All participants were healthy, mobile, and physically active, further confirming intact motor function. These details have been added to the methods section (lines 135–144).

3. It would be interesting, in addition to the sample morphologic characteristics, to describe their level of cognitive (eg students) and physical activity (sedentary) if you collected them.

Response: The study participants were undergraduate and graduate students, ensuring a cognitively active sample. Yes, we assessed their physical activity levels using specific questions about the type and intensity of their usual physical activities as below:

• Light activity: Activities where the heart beats slightly faster than normal, and participants can talk or sing while performing them (e.g., leisure walking, stretching, light yard work, or vacuuming).

• Moderate activity: Activities where the heart beats faster than normal, participants can talk but not sing, and examples include fast walking, aerobics, strength training, or gentle swimming.

• Vigorous activity: Activities where the heart rate increases significantly, and talking is difficult or broken up by large breaths.

All participants reported engaging in either light, moderate, or vigorous physical activities. We included this information in the participants’ section. Please see lines 144–145.

4. Please give an additional explanation of why complex reaction time tasks are more resource-demanding for cortical areas.

Response: Complex reaction time task (CRT) necessitates the coordination of several cognitive functions, including working memory to retain sound patterns and their corresponding responses, sustained attention to maintain focus, auditory discrimination to identify pitch variations, decision-making to select the appropriate response, and executive action to execute the motor response (McGuire et al., 2010). The PFC is critically engaged during this multifaceted process. A growing body of research indicates that PFC activation is significantly heightened during high-demand tasks compared to low-demand counterparts (McGuire et al., 2010; Ohsugi et al., 2013; Liang et al., 2016). This augmented neural activity is often accompanied by an increase in cerebral blood volume, which serves as a proxy for neuronal activation and reflects the metabolic demands associated with task performance.

We included this explanation and below references in the manuscript. Please see lines 194 – 202.

J.T. McGuire, M.M. Botvinick, Prefrontal cortex, cognitive control, and the registration of decision costs, Proc. Natl. Acad. Sci. U.S.A. 107 (17) 7922-7926, ttps://doi.org/10.1073/pnas.0910662107 (2010).

Ohsugi, H., Ohgi, S., Shigemori, K. et al. Differences in dual-task performance and prefrontal cortex activation between younger and older adults. BMC Neurosci 14, 10 (2013). https://doi.org/10.1186/1471-2202-14-10

Liang, L. , Shewokis, P. and Getchell, N. (2016) Brain Activation in the Prefrontal Cortex during Motor and Cognitive Tasks in Adults. Journal of Behavioral and Brain Science, 6, 463-474. doi: 10.4236/jbbs.2016.612042.

Results

5. Please use international metric Kg and m.

Response: Suggested changes have been implemented using international metric units (kg and m). Please see Table 1.

6. Place the table 1 heading on top of it.

Response: The suggested changes have been implemented, and the Table 1 heading has been moved to the top of the table. Please see line 310.

Reviewer #2: The authors present results of a single-arm study assessing 5-days of dual-task training and its potential impact on dual-task and single task performance the day after training ends and 1-week post-training. The authors also assess changes in cortical activation through fNIRS in the prefrontal cortex (PFC) and vestibular cortex. Authors show improvement in balance during training and improvements in task performance (both single and dual-task) immediately post training and at follow-up. There are also changes observed in cortical activation. The manuscript will be strengthened if the authors address the following points.

1. Authors should clarify the meaning of "trials" and "blocks". It seems that the meaning may change in different sections of the paper which causes some confusion. For example, line 201 says 6 trials with one trial per block but line 209 indicates 3 blocks of six trials, Figure 4 refers to each component as a block and line 222 says each trial had a duration of 210 seconds, while a block was 30 seconds.

Response: Thank you for your thoughtful feedback. We agree that the simultaneous use of "trials" and "blocks" could lead to confusion. To enhance clarity, we have removed the term "block" and consistently used "trials" throughout the manuscript. In the original version, a "block" referred to a set of six trials, followed by a two-minute rest to reduce fatigue. Please see lines 225, 233 – 235, 241 – 245.

2. Authors should clarify their analytic approach for the fNIRS data including the multiple comparison correction strategy. Authors state using both FDR and Bonferroni correction. In lines 253-261, authors mention linear mixed effects models, but then later (lines 287) they talk about mixed model ANOVA and then say a p-value less than 0.05 is considered significant, when earlier they have lower cut-offs presented. I wonder if it would make sense to just describe the methods of analysis just in the statistical analysis section...

Response: We have revised the fNIRS signal processing and statistical analysis sections for clarity. During the fNIRS signal processing, a linear mixed model is used to identify the correlations between the ROIs and the changes in the fNIRS signals, which is not for the final results. We have separated the stats for fNIRS signal processing and stats for fNIRs data comparisons. Please see lines 273 – 281 for fNIRS signal processing and lines 307 – 312 for fNIRS data comparisons. T-tests with the Benjamini-Hochberg procedure and Bonferroni correction were conducted for multiple comparisons to reduce the complexity of the statistical model and avoid Type I errors.

3. Authors use various forms of ANOVA for analysis. Were the assumptions of these methods met by the data?

Response: The assumption of normality was met. The data was normally distributed; hence, the ANOVA was used. We included these details in the manuscript. Please see line 301.

4. Authors need to be careful in their statements about their findings. Although they acknowledge in the limitations that there is no control group, which limits interpretation of causality, much of their phrasing throughout the manuscript suggests causality. Authors do not know what might have happened to cortical activation or even performance on tasks at the different time points without the training.

Response: Thank you for your valuable feedback. We have carefully revised the phrasing throughout the discussion to avoid suggesting causality and to ensure our interpretation aligns with the limitations of the study.

Minor points

1. Line 42: authors state that DT training led to decreased activation in the bilateral PFC, but this was only true (in the dual-task) for balance+complex reaction time.

Response: We modified the sentence to reflect those changes pertaining to complex dual-task condition. Please see lines 42- 43.

2. Figure 2: I suggest adding the timing of visits 7 and 8 to the figure (visit 7 the day after training ends and visit 8 1-week later) so that readers know they are not all consecutive days without reading the text.

Response: We made the suggested changes in figure 2. Please see revised Figure 2.

3. Would the protocol for the reaction time tasks changed if individuals were left-handed? I realize all participants were right-handed in this study, but it might be worth stating that in the description of the reaction time task section (lines 169-181), so a reader knows that (especially if it would affect the protocol if someone was actually left-handed). Currently, the reader doesn't know everyone is right-handed until Table 1.

Response: Yes, the placement of clicker would have changed for the left-handed person. In the simple reaction time task (SRT), participants would have used a single clicker held in their left (dominant) hand to respond to a 500 Hz tone delivered at 60% volume through in-ear headphones. In the complex reaction time task (CRT), participants would have responded to the 500 Hz tone with the clicker in the left hand, while to the 1000 Hz tone with a separate clicker held in the right (non-dominant) hand.

As we had all right-handed young adults, to clarify and align the methodology with right hand dominant participant, we included dominant and non-dominant corresponding to the right and left hand respectively in the reaction times methodology. Please see lines 190 – 193.

4. line 221: change "30 second" to "30 seconds"

Response: We made the suggested changes. Please see line 245.

5. Table 1: I would include the range of ages, even though the data are provided in supplemental material (especially since the inclusion criteria had a much wider age range than what was ultimately used).

Response: We made the suggested changes. Please see table 1.

6. Table 2 -for the complex reaction time section, change "Time vs Time" to "Visit vs Visit" to be consistent with the simple reaction time section

Response: We made the suggested changes. Please see table 2.

7. Figure 6: there is a typographical error in the row labels for the balance+SRT (change "Performace" to "Performance"). Also, define the error bars and * in the figure caption. Is the bar height the observed mean at the different time points? This should be clarified. Authors might also consider a more informative plot, especially given the small sample size (box-plots with individual data points overlaid) - and consider that same type of plot for other figures that are presenting observed means (if that is what is presented).

Response: We sincerely appreciate your feedback and have made the suggested corrections to address the typographical error. The bars in the figures represent the means, and the error bars denote the standard deviations; this has been clarified in the figure captions (Please see line 359). Regarding the use of bar graphs, we chose to retain them over box plots with individual data points, as the bar graphs appeared to provide less cluttered visualization and offer a more compelling narrative for the results. Additionally, bar graphs are commonly used for certain outcomes, such as fNIRS data, ensuring consistency with established practices.

8. lines 351-352 - authors have switched SRT and CRT. RT was longer for CRT not SRT.

Response: We made the suggested changes. Please see lines 373 – 374.

9. Figure 8: row labels - change "testing" to "training". Also define the error bars and *.

Response: We made the suggested changes. Please see revised figure 8 and the caption.

10. Figure 9 caption H is listed twice - the second one should be "J".

Response: We made the suggested changes. Please see line 392.

11. All references to figures in lines 374 through the end of the results should be changed from Figure 6 to Figure 9

Response: We made the suggested changes. Please see lines 400 – 418.

---

## [Decision Letter · Decision Letter 1]

16 Mar 2025

Effects of dual-task training on cognitive-motor learning and cortical activation: a non-randomized clinical trial in healthy young adults

PONE-D-24-40678R1

Dear Dr. Surkar,

We’re pleased to inform you that your manuscript has been judged scientifically suitable for publication and will be formally accepted for publication once it meets all outstanding technical requirements.

Kind regards,

Stephen D. Ginsberg, Ph.D.

Section Editor

PLOS ONE

**Comments to the Author**

1. If the authors have adequately addressed your comments raised in a previous round of review and you feel that this manuscript is now acceptable for publication, you may indicate that here to bypass the “Comments to the Author” section, enter your conflict of interest statement in the “Confidential to Editor” section, and submit your "Accept" recommendation.

Reviewer #2: All comments have been addressed

2. Is the manuscript technically sound, and do the data support the conclusions?

Reviewer #2: (No Response)

3. Has the statistical analysis been performed appropriately and rigorously? 

Reviewer #2: (No Response)

4. Have the authors made all data underlying the findings in their manuscript fully available?

Reviewer #2: (No Response)

5. Is the manuscript presented in an intelligible fashion and written in standard English?

Reviewer #2: (No Response)

6. Review Comments to the Author

Reviewer #2: (No Response)

7. PLOS authors have the option to publish the peer review history of their article (what does this mean? ). If published, this will include your full peer review and any attached files.

**Do you want your identity to be public for this peer review?** For information about this choice, including consent withdrawal, please see our Privacy Policy .

Reviewer #2: No

---

## [Editor Report · Acceptance letter]

PONE-D-24-40678R1

PLOS ONE

Dear Dr. Surkar,

I'm pleased to inform you that your manuscript has been deemed suitable for publication in PLOS ONE. Congratulations! Your manuscript is now being handed over to our production team.

Kind regards,

on behalf of

Dr. Stephen D. Ginsberg

Section Editor

PLOS ONE